# Aquatic Macroinvertebrate Indicators in the Zawgyi Irrigation Channels and a River in the Central Dry Zone of Myanmar

**Nyein Thandar Ko** [1,*] , **Phil Suter** [2] , **John Conallin** [3,4] , **Martine Rutten** [1] **and Thom Bogaard** [1]

1   Department of Water Management, Delft University of Technology, Stevinweg 1, 2628CN Delft, The Netherlands; m.m.rutten@tudelft.nl (M.R.); T.A.Bogaard@tudelft.nl (T.B.)
2   Department of Ecology, Environment and Evolution, La Trobe University, Wodonga, VIC 3690, Australia; p.suter@latrobe.edu.au
3   Institute of Land Water and Society Charles Sturt University, Elizabeth Mitchell Dr., Albury, NSW 2640, Australia; jconallin@csu.edu.au
4   Department of Water Resources and Ecosystems, IHE Delft, Westvest 7, 2611 AX Delft, The Netherlands
*   Correspondence: N.T.Ko@tudelft.nl

**Abstract:** Rivers and wetlands in Myanmar provide essential services to people in terms of transportation, agriculture, fisheries and a myriad of other ecosystem services, all of which are dependent on a healthy ecosystem. Irrigation channels are also an important part of the infrastructure for daily water use in Myanmar. The objective of this research is to describe the aquatic ecosystem of irrigation channels using aquatic macroinvertebrate communities. The research focused on the taxonomic composition of the aquatic macroinvertebrates of the Zawgyi River and the associated irrigation channels in central Myanmar, east of the city of Mandalay. Significant differences between the river and channels, and among individual channels, were shown using an analysis of similarity: Bray–Curtis similarity, a multivariate equivalent of the univariate statistical method of analysis of variance: ANOSIM and an analysis of similarity percentages: SIMPER by Plymouth Routines in Multivariate Ecological Research: PRIMER v6 software. The initial findings suggest that there is a clear separation between macroinvertebrate communities at the morpho-species level of identification between river and irrigation channels, while there is less separation between functional feeding groups (FFG) between them. The lower taxonomic level of discrimination at the family level using a water quality index showed no significant difference between river and channels. The preliminary field results indicate that a recently modified biomonitoring index method could be applied in Myanmar to assess the ecological water quality of the modified river, as well as human-made channels.

**Keywords:** irrigation channels; macroinvertebrates; functional feeding groups (FFG); water quality index

## 1. Introduction

A better understanding of aquatic biodiversity values in different water body types is vital to achieve sustainable freshwater ecosystems [1,2]. However, there is limited research on the comparison of freshwater aquatic biodiversity values within different water bodies (e.g., rivers, streams, lakes, ponds, and ditches), especially regarding novel ecosystems such as human-contrived channels and ditches [1,3,4]. Traditional aquatic ecological research has focused exclusively on larger water bodies, larger rivers, streams and lakes [5]. Smaller water bodies, such as channels, ditches and ponds, have recently received growing interest due to their abundance, importance for freshwater biodiversity and global biogeochemical cycles [6,7]. Recent studies have concentrated on natural smaller water

bodies (e.g., streams and springs) based on aquatic communities and diversity [8–10]. Nonetheless, the aquatic biodiversity characteristics of semi-natural and artificial smaller water bodies (irrigation channels and ponds) remain understudied [7,11].

In practice, artificial or human-made smaller water bodies, such as irrigation channels, are important components of farming infrastructure for human society and can also provide a habitat for aquatic flora and fauna communities, with many having distinct aquatic species of macroinvertebrates present [12]. In general, the principal purpose of human-made channels is agricultural, where they serve as conduits to deliver agricultural water and remove storm-water runoff [11]. Further, extensive irrigation channels replace the natural headwaters of regional watersheds [12] and serve as an interface between agriculture and aquatic ecosystems [13]. Alternatively, irrigation channels are significantly relevant to the optimization of productive agricultural areas globally [13], especially in Southeast Asia (SE Asia), which accounts for 60% of the world's irrigated area [14].

Myanmar is one of the most agriculturally dominated countries in SE Asia. People's income and the country's economy mainly depend on the availability of water: the agricultural sector employs more than 65% of the population [15]. Approximately one million hectares of the irrigated area traditionally receive water from the irrigation channels through weirs and dams established by about 200 irrigation projects [16]. Lazarus et al. [17] reported that there are around 180 irrigation reservoirs used for irrigation water supply. Irrigation channels are the most common waterways throughout urban and rural areas, with daily water use by the population reliant on them, but pollution sources are of concern in these systems.

The surface water in channels is susceptible to anthropogenic activities, including pollution from urban wastewater, stormwater runoff and the use of synthetic fertilizer and pesticides [18]. Polluted irrigation water causes negative impacts on food production, water quality [3], aquatic communities, faunal diversity and human health [4]. Irrigation channels may be a habitat that macroinvertebrates can occupy, but they are artificial and do not offer habitats that are as diverse as natural streams and therefore do not support biodiversity in an equivalent manner to streams. Similar to previous studies, the authors in [5,11,12] found that irrigation channels do not have the same aquatic taxonomic richness as streams, rivers, lakes and ponds due to the different environmental features associated with them. On the contrary, some studies have shown that irrigation channels in lowland fens and ancient wetlands sometimes support higher taxonomic richness than the larger downstream rivers [2,5,11,19].

Macroinvertebrate communities have emerged as an appropriate representative of aquatic biodiversity communities to provide a time-integrated measurement of freshwater ecosystem characteristics [20]. Changes in an environmental gradient within water bodies can influence the variation of the richness of aquatic taxa and their abundance due to the availability of different food sources (e.g., organic matter, nutrient resources) [21]. There are five functional feeding groups [22]: shredders (which consume leaf debris–coarse particulate organic matter (CPOM)), collectors (which consume deposited fine-particulate organic matter (FPOM)), filterers (which consume suspended fine detritus and organic material), grazers (who consume algae) and predators (which consume other invertebrate prey). These different functional feeding groups can be used to detect the health of the function of the freshwater ecosystem [21].

To date, there is little knowledge about the aquatic biodiversity of natural water bodies in Myanmar. Recently, Ko et al. [23] reported on the development of the first Myanmar-based biomonitoring assessment tool to detect the ecological water quality index. This tool has been developed and tested using macroinvertebrate samples from the least impacted or minimally impacted sites of the Myitnge and Chaumagyi Rivers, east to northeast of Mandalay, Myanmar. In this region, large-scale irrigation works exist; however, to date, the aquatic ecosystem of these irrigation channels has not been assessed.

The objective of this work was to compare the aquatic ecosystems of irrigation channels with the river using macroinvertebrate communities. We describe the ecosystems of irrigation channels and associated rivers in terms of their taxa richness, abundance, functional feeding groups and different land uses. We performed sampling in irrigation channels within the vicinity of urban settlements and a rural area with intensive and extensive agriculture, as well as in a natural river (the Zawgyi River). We further applied the first Myanmar-based biomonitoring index [23] to determine the current status of ecological water quality in both the river and channels. With this approach, we investigated the suitability of the method for the impacted sites of the river, as well as for human-made irrigation channels.

## 2. Materials and Methods

### 2.1. Study Area

We selected the Zawgyi Irrigation Network located around the Kyaukse township, 40 km south of Mandalay city in the central dry zone of Myanmar. Kyaukse is a town within the irrigation network of the Zawgyi River, with an estimated population of 257,907 in 2014 [24].

The Zawgyi River flows through the Shan Plateau of eastern Myanmar and then joins the Myitnge River. There is a dam (Zawgyi dam) $21°33'52.13''$ N, and $96°52'25.31''$ E upstream on the Zawgyi River in Shan state. At 100 km below the Zawgyi dam, there is a second dam, namely Myogyi dam (Figure 1). The governmental Irrigation Department constructed four diversion weirs at 13 km, 24 km, 41 km and 41.5 km downstream of Myogyi dam, respectively. Six irrigation channels were built along this stretch of the river, and these four glacis weirs around the Kyaukse area are used to provide water for agricultural land and urban daily water use.

The size of the channels varies from a small ditch with a depth of 1 m and a width of 6 m to larger ditches with a depth of 1.8 m and a width of 31 m. The capacity of the ditches ranges from $2.8\,\mathrm{m^3\,s^{-1}}$ to $27.4\,\mathrm{m^3\,s^{-1}}$, and their service area ranges from 16.82 $\mathrm{km^2}$ to 104.78 $\mathrm{km^2}$ based on the size and capacity of the channels. We selected both the Zawgyi River and its six irrigation channels to collect macroinvertebrates samples and in situ physico-chemical water quality parameters (Figure 1).

### 2.2. Physico-Chemical Water Quality Variables

We collected five in situ physico-chemical parameters (pH, turbidity (Nephelometric Turbidity Units, NTU), surface water temperature (Temp) [°C], electrical conductivity (EC) ($\mathrm{\mu S\,cm^{-1}}$) and water velocity ($\mathrm{m\,s^{-1}}$) at all sampling sites in both the Zawgyi River and irrigation channels at the time of sampling. We used Simplex Health pH test strips with a range from 4.5 to 9.0, Ground Truth water clarity tubes, Greisinger, a GMH 3400 series meter and a Transparent Velocity Head Rod (TVHR) (http://www.groundtruth.co.za/our-products/).

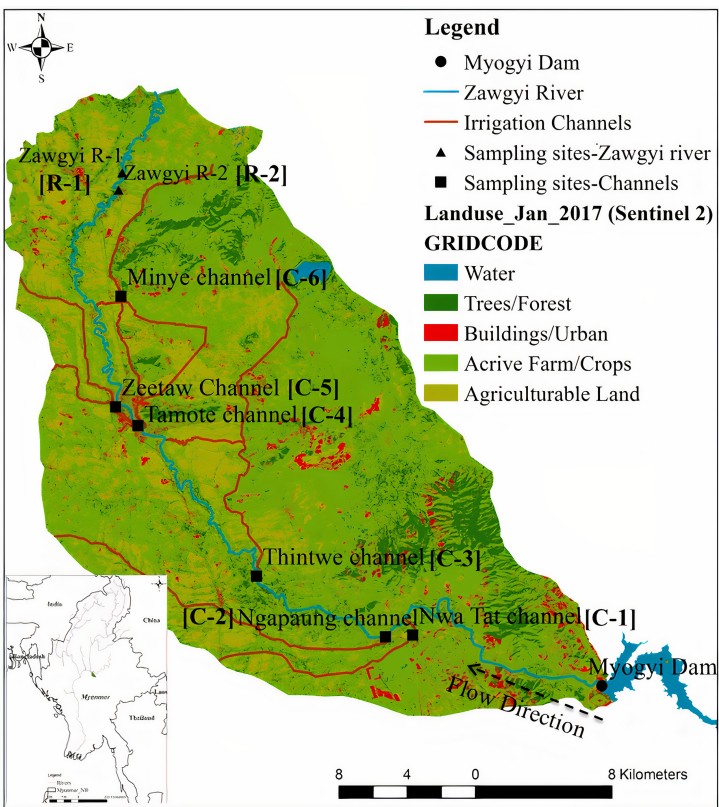

**Figure 1.** Location of sampling sites on the Zawgyi River and its irrigation channels and the dam on the Zawgyi River on a map showing different land uses, based on Sentinel 2 imagery; R: river; C: channel.

*2.3. Macroinvertebrate Sampling*

We collected macroinvertebrate samples from six irrigation channels (C-1, C-2, C-3, C-4, C-5 and C-6) and two sampling sites in the Zawgyi River (R-1 and R-2) during the period from November to December 2016 (Figure 1). All channel and river sites are impacted by human modifications and are located in an agricultural landscape. We repeated sampling in two sites of the Zawgyi River (R-II-1 and R-II-2) between February and March 2017, but we could not repeat sample collection in the irrigation channels as there was a high water level due to unexpected heavy rainfall. The sampling sites on the river lie at the downstream of the Myogyi dam, as well as downstream of rural and urban settlements before the Zawgyi River joins the Myitnge River.

Both sampling sites in the Zawgyi River have a rocky substrate with vegetation along the riverbank, while all sampling sites in the channels have an earthen or clay-based substrate without vegetation along the edge. Sites C-1, C-2 and C-3 were located in an area with agricultural farming (agricultural). Sites C-4 and C-5 were in the city centre of Kyaukse (urban). Sample site C-6 was outside of the city and located in a rural settlement (rural) (Figure 1).

We collected aquatic macroinvertebrate communities from the bank of each channel and the Zawgyi River using a sweep-sample technique with nylon nets (500 micron mesh) [8]. Three individual sweeps each—over a sampling length of 5 m parallel to the bank—were taken from each site. A total of 30 macroinvertebrate samples from the river and channels were collected in the period between December 2016 to March 2017. Specimens were preserved in 70% ethanol in the field. We identified all collected Arthropoda (insects and crustaceans) and Mollusca (gastropod snails and bivalves) as their family and species in the laboratory using keys from [22,25]. Feeding traits for each taxon were determined based on the functional feeding groups (FFG) in Dudgeon [22] and Merritt and Cummins [26].

## 2.4. Ecological Water Quality Index

We applied the first Myanmar-based biomonitoring index developed by Ko et al. [23] to determine the current status of ecological water quality in both the river and channels based upon information from all macroinvertebrte taxa present in each site. The first Myanmar-based biomonitoring index method was developed by modifying the index of the Asia Foundation biomonitoring method, which was developed from the Mekong River basin. This revised standard scoring method represents the first index method in Myanmar and includes 48 taxonomic groups. The average score per taxon (ASPT) of this method ranges from 1 to 10 based on tolerant to intolerant water quality. We calculated the ecological water quality index of the river and channels with the following equation by Ko et al. [23]:

$$\text{Ecological Water Quality Index} = \frac{\text{Sum of ASPT}}{\text{Total number of taxa}} \tag{1}$$

Ko et al. [23] defined three ecological water quality classes based on the calculated index: a value between 1 to 2.9 means a poor or a largely to seriously modified ecosystem by human activities with serious deviation from the natural ecosystem, a score between 3.0 and 6.9 means a fair or moderately modified ecosystem by human activities with little deviation from the natural ecosystem, and a score between 7.0 and 10.0 means a good or unmodified to little-modified condition by human activities.

## 2.5. Statistical Analysis

Statistical analysis was performed using the statistical program PRIMER v6 [27]. We tested different statistical variables: for all present taxa, for only Ephemeroptera, Plecoptera, Trichoptera and Odonata (EPTO) and for FFG based on abundance and species contributions, in order to compare the community difference between different water bodies, we computed the resemblance matrix by using Bray–Curtis similarity using a square root transformation, then applied non-metric multidimensional scaling (MDS) for visualization to all resemblance matrixes by using Bray–Curtis similarity with 999 random stars. An MDS plot shows the stress value, which represents the rank order correlation distance among the resemblance matrix. A stress value of less than 0.05 means a near-perfect representation of ordination space among samples [28].

If there was any significant difference between samples (variables), we used a one-way analysis of similarity (ANOSIM) to confirm clear separation based on a null hypothesis of no community/FFG/index differences between channels and between the river and irrigation channels. ANOSIM calculates a test statistic (Global R between 0 to 1) that provides a comparative measure of the degree of separation of pre-defined sampling groups and its probability of occurring by chance using 999 permutations [27]. The higher the Global R-value, the larger the differences between samples, but if all permuted statistics are higher than the Global R, then the null hypothesis can be rejected at 0.001 significance [27].

Additionally, we used an analysis of similarity percentages (SIMPER) to determine the percentage of (dis)similarity of macroinvertebrate communities among irrigation channels and between the river and irrigation channel sites and the contribution of individual taxa to the resulting dissimilarity. In SIMPER analysis, we used the feeding traits of the macroinvertebrates as a variable to examine the contribution of each feeding behavior between the two water bodies.

## 3. Results

### 3.1. Physico-Chemical Water Quality

Figure 2 shows the collected physico-chemical attributes of the irrigation channels and the Zawgyi River. The values of pH ranged from 7.25 to 8.0. The turbidity values of sites in the river were higher than the values in some sampling sites of the channel (C-1, C-2, C-3 and C-6) but similar to the value in C-4 and C-5. We obtained a higher EC in the river (range of 547–550 $\mu S\,cm^{-1}$) than in the channels

(range of 329–473 $\mu$S cm$^{-1}$). The flow velocity at the time of sampling ranged from 0.12 m s$^{-1}$ to 0.92 m s$^{-1}$ in the channels and from 0.3 m s$^{-1}$ to 0.5 m s$^{-1}$ in the river.

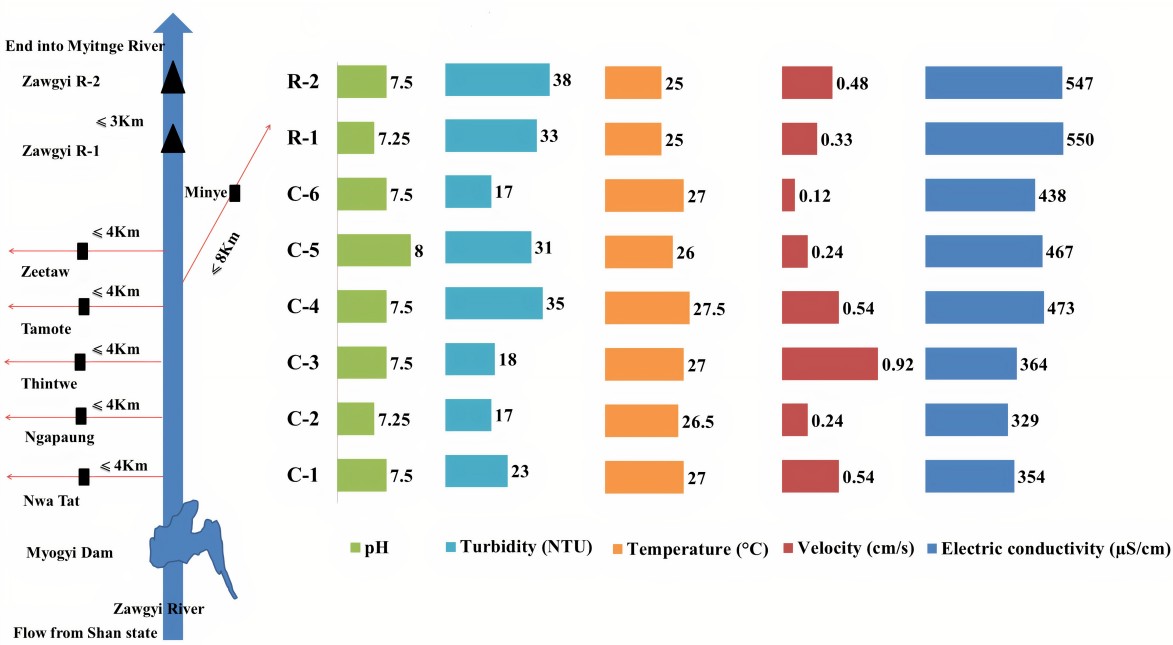

**Figure 2.** Schematic diagram with the distance from the main river to sampling sites on irrigation channels with the collected physico-chemical parameters between November and December 2016. NTU: Nephelometric Turbidity Units.

### 3.2. Qualitative Comparisons of Taxa Richness, Abundance and Taxa Composition

Table 1 shows all collected aquatic invertebrates at the order, family and morpho-species-level taxonomies from all sampling sites of the river and channels, as well as their representative FFG and ASPT. A total of 22 aquatic invertebrate taxa from 18 families in 10 different orders were recorded from all samples in the river and channel sites (Table 1). The channels had lower diversity than the river sites, with 15 taxa from 13 families in six orders in the channels, while there were 19 taxa from 16 families in 10 orders in the river. Sample site R-II-2 on the Zawgyi River showed the highest abundance, while site C-6 showed the lowest abundance. The highest richness was 13 taxa in site C-4, and the lowest richness was three taxa in site C-6 (Table 1).

Aquatic insects dominated the majority of taxa at all sampling sites, with 11 taxa in channels and 13 taxa in the river. The most common insect order in both channels and the river was Ephemeroptera, with three families: Baetidae, Heptagenidae and Caenidae. Of these, Caenidae were most abundant, ranging from 4 to 65 individuals in the channels and from 1 to 2072 individuals in the river. Our sampling recorded three non-insect orders, with four and six taxa from the channels and the river, respectively. All four main functional feeding groups—collectors, filterers, grazers and predators—were present in both the river and channels.

**Table 1.** Collected macroinvertebrate communities at the order, family and morpho-species level of the Zawgyi River and its six irrigation channels (L: larvae, A: adults, P: pupae, sp: species). The Average Score per Taxon (ASPT) is taken from the first Myanmar Index method by Ko et al. [23]. FFG: functional feeding group.

| Order | Family | Species | FFG | ASPT | Irrigation Channels | | | | | | Zawgyi River | | | |
| | | | | | C-1 | C-2 | C-3 | C-4 | C-5 | C-6 | R-1 | R-2 | R-II-1 | R-II-2 |
|---|---|---|---|---|---|---|---|---|---|---|---|---|---|---|
| Mesogastropoda | Bithynidae | Bithynidae sp. 1 | Grazer | 6 | 0 | 0 | 0 | 0 | 0 | 0 | 0 | 4 | 0 | 0 |
| | Thiaridae | Thiaridae sp. 1 | Grazer | 6 | 16 | 0 | 11 | 11 | 0 | 22 | 0 | 7 | 3 | 5 |
| Bivalvia | Corbiculidae | *Corbicula* sp. 1 | Filterer | 3 | 7 | 0 | 0 | 21 | 46 | 0 | 33 | 2 | 1 | 0 |
| | Margaritiferidae | Margaritiferidae sp. 1 | Filterer | 3 | 0 | 0 | 0 | 1 | 0 | 0 | 0 | 1 | 0 | 0 |
| Decapoda | Palaemonidae | *Macrobrachium* sp. 1 | Predator | 8 | 0 | 0 | 0 | 0 | 0 | 0 | 0 | 17 | 0 | 0 |
| | Parathephusidae | Parathelphusidae sp. 1 | Collector | 3 | 0 | 1 | 0 | 1 | 0 | 0 | 2 | 0 | 1 | 0 |
| Ephemeroptera | Baetidae | *Baetis* sp. 1 | Collector | 5 | 8 | 5 | 19 | 15 | 0 | 0 | 78 | 36 | 76 | 22 |
| | | *Platybaetis* sp. 1 | Collector | 5 | 0 | 0 | 0 | 0 | 0 | 0 | 17 | 3 | 65 | 0 |
| | Heptagenidae | *Asionurus* sp. 1 | Filterer | 10 | 4 | 0 | 5 | 16 | 0 | 0 | 1 | 0 | 0 | 0 |
| | | *Asionurus* sp. 2 | Filterer | 10 | 0 | 0 | 0 | 2 | 0 | 0 | 0 | 0 | 0 | 0 |
| | Caenidae | *Caenis* sp. 1 | Collector | 4 | 25 | 17 | 40 | 65 | 4 | 9 | 25 | 1 | 1689 | 2072 |
| Odonata | Gomphidae | Gomphidae sp. 1 | Predator | 6 | 3 | 9 | 2 | 5 | 0 | 0 | 0 | 2 | 0 | 0 |
| Hemiptera | Micronectidae | Micronectidae sp. 1 | Predator | 1 | 0 | 0 | 0 | 0 | 0 | 0 | 0 | 1 | 0 | 0 |
| Trichoptera | Hydropsychidae | *Potamyia* sp. 1 | Filterer | 5 | 0 | 0 | 0 | 0 | 0 | 0 | 50 | 5 | 3 | 7 |
| Coleoptera | Dytiscidae | Dytiscidae sp. 1 | Predator | 2 | 2 | 50 | 0 | 0 | 4 | 0 | 0 | 0 | 0 | 0 |
| | Elmidae | Elmidae (L) | Predator | 8 | 0 | 0 | 0 | 2 | 0 | 8 | 1 | 0 | 0 | 31 |
| | | Elimidae (A) | Predator | 8 | 1 | 0 | 0 | 1 | 0 | 0 | 1 | 0 | 0 | 0 |
| | Hydrophilidae | Hydrophilidae sp. 1 | Collector | 3 | 2 | 23 | 0 | 0 | 1 | 0 | 0 | 0 | 0 | 0 |
| Diptera | Chironomidae | Tanypodinae sp. 1 | Predator | 3 | 0 | 0 | 0 | 3 | 4 | 0 | 0 | 0 | 45 | 85 |
| | | Chironomidae (P) | | 3 | 0 | 0 | 0 | 0 | 0 | 0 | 1 | 0 | 20 | 23 |
| | Ceratopogonidae | Ceratopogonidae sp. 1 | Predator | 3 | 0 | 0 | 0 | 4 | 4 | 0 | 0 | 0 | 1 | 1 |
| Neuroptera | Sisyridae | Sisyridae sp. 1 | Predator | 3 | 0 | 0 | 0 | 0 | 0 | 0 | 0 | 0 | 0 | 1 |
| Total species richness | | | | | 9 | 6 | 5 | 13 | 6 | 3 | 10 | 11 | 10 | 9 |
| Total abundance | | | | | 68 | 105 | 77 | 147 | 63 | 39 | 209 | 79 | 1904 | 2247 |

*3.3. Comparisons of Macroinvertebrate Communities at Morpho-Species Level between the River and Channels*

Based on Bray–Curtis similarity using all aquatic taxa at the lowest taxonomic level (morpho-species), there was a separation between the two water body types. The MDS ordination plot showed a significant separation between the communities in the river and channels (stress = 0.17) (Figure 3a). ANOSIM also indicated that there was no overlap between the macroinvertebrate assemblage between the river and channels (R statistic = 0.489), with a dissimilarity between these two water bodies of 80% (Table 2).

The percentage of average dissimilarity in the community decreased to 64% when only taxa (Ephemeroptera, Plecoptera, Trichoptera, Odonata and EPTO) at the morpho-species level were considered (Table 2), and the stress level of MDS ordination was also lower (stress = 0.12) (Figure 3b). Table 2 shows the lists of results of SIMPER and ANOSIM for comparisons between the river and channels using different variables. The comparison of the river and channels using FFG showed less separation than the other two comparisons using all taxa and EPTO (Table 2).

Besides this, the difference in FFG composition between the river and channels based on different land uses (urban, rural, agricultural) showed that there was no significant difference between different land uses in the present study (Table 2). However, the comparison of macroinvertebrate communities based on FFG composition between the open river and channels in urban groups showed over 70% average dissimilarity.

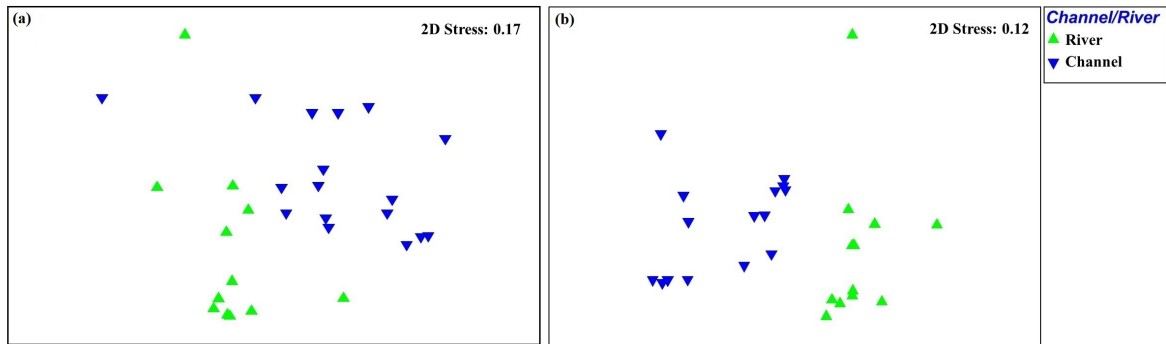

**Figure 3.** Dimensionless multidimensional scaling (MDS) ordination plot for the comparison of the communities of the Zawgyi River and six irrigation channels: (**a**) MDS plot using all taxa at the morpho-species level, (**b**) MDS plot using only Ephemeroptera, Plecoptera, Trichoptera and Odonata (EPTO) taxa at the morpho-species level.

**Table 2.** Results of an analysis of similarity percentages (SIMPER) (average dissimilarity) and one-way analysis of similarity (ANOSIM) (Global statistic R, significant level of statistic, *p*-value) for the comparison of macroinvertebrate communities based on all taxa at the morpho-species level, tolerant taxa (Ephemeroptera, Plecoptera, Trichoptera, Odonata, EPTO) at the morpho-species level and functional feeding groups (FFG) of morpho-species contributions between the Zawgyi River and six Irrigation Channels

| Waterbodies | Variable | (SIMPER) Average Dissimilarity [%] | ANOSIM R-Statistic | *p*-Value |
|---|---|---|---|---|
| | All taxa | 80 | 0.489 | 0.001 |
| River and Channels | EPTO | 64 | 0.645 | 0.001 |
| | FFG | 61 | 0.303 | 0.003 |
| River--Urban channels | | 58 | 0.136 | 0.110 |
| River--Rural channel | FFG | 73 | 0.531 | 0.004 |
| River--Agricultural channels | | 58 | 0.262 | 0.012 |

Note: Urban channels are C-4 and C-5; the rural channel is C-6; agricultural channels are C-1, C-2, and C-3.

Figure 4 shows the bubble plots of the number of individuals of different FFG at each sampling site. In four FFG groups, collectors were high in abundance in all sampling sites of the river, except for site R-2, and were absent from C-5 and rare in C-4 and C-5 (Figure 4a). Filterers were rare in sampling

sites of the river (R-2, R-II-1, R-II-2) in comparison with the sites in channels C-1, C-2, C-4 and C-5 (Figure 4b). We found a high abundance of filterers only in the river site (R-1) (Figure 4b). Grazers had high abundance in site R-1 but were absent in sites C-4, C-5 and C-2 (Figure 4c). The abundance of predators was highest in river sites and sites C-2 and C3, but these were rare or absent at all other sites (Figure 4d).

Table 3 shows the average dissimilarity percentage and FFG with the highest contribution in percentage between the sites of the river and six channels. Collectors were the dominant FFG group to control all dissimilarity percentages. The highest average dissimilarity percentage was found between C-5 and sites R-1 and R-2, with over 70% in both sites of the river (Table 3).

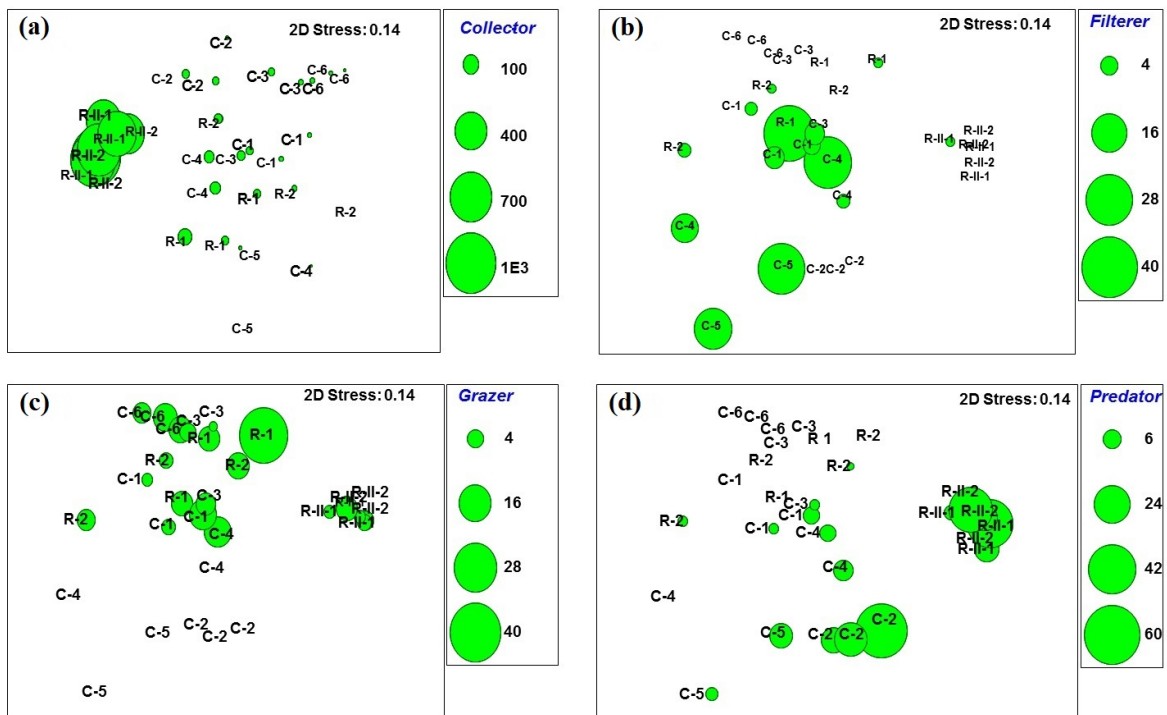

**Figure 4.** Bubble plot of present abundance by dimensionless MDS ordination based on four FFG (**a**): Collectors, (**b**): Filterers, (**c**) scraper grazers and (**d**) predators with their respective abundance numbers between the Zawgyi River (R) and irrigation channels (C).

**Table 3.** Average dissimilarity percentages of each sampling site of the river and channels with the respective functional feeding group (FFG) of the highest species contribution and their contribution (%) using functional feeding groups (FFG) by an analysis of similarity percentages (SIMPER).

| Channels | River | Average Dissimilarity [%] | FFG of the Highest Species Contribution | Contribution [%] |
|---|---|---|---|---|
| C-1 | R-1 | 54 | Collector | 63 |
| | R-2 | 52 | Collector | 68 |
| C-2 | R-1 | 70 | Collector | 54 |
| | R-2 | 66 | Collector | 56 |
| C-3 | R-1 | 50 | Collector | 64 |
| | R-2 | 54 | Collector | 69 |
| C-4 | R-1 | 57 | Collector | 56 |
| | R-2 | 62 | Collector | 57 |
| C-5 | R-1 | 77 | Collector | 57 |
| | R-2 | 77 | Collector | 52 |
| C-6 | R-1 | 60 | Collector | 71 |
| | R-2 | 59 | Collector | 70 |

### 3.4. Comparison of Macroinvertebrate Communities at the Morpho-Species Level in the Irrigation Channels

The resemblance matrix of all macroinvertebrate assemblages at the morpho-species level among the six different channels obtained with Bray–Curtis similarity generated a clear separation among samples, which is displayed on the two-dimensional MDS ordination plot (stress = 0.13) (Figure 5a). The MDS plot shows that there was no overlap between macroinvertebrate communities within the six channels. The Global R statistic was high (0.714), with a significantly low level of the sample statistic ($p$ = 0.001), demonstrating that there was a good separation in the communities between different channels (Figure 5b).

The similarity analysis using SIMPER provided the average percentage of (dis)similarity among the individual channels. Table 4 shows the results of SIMPER with the highest species contribution alongside the contribution percentages and FFG for each group for all taxa and individual taxa (EPTO). The highest average dissimilarity percentages were found in the groups of C-5 and C-6, at 91%, while the lowest average dissimilarity percentages were shown in the groups of C-1 and C-3, with 40%, when all taxa are considered. The highest taxa contribution of groups C-5 and C-6 were the filter-feeding bivalves Mollusca–*Corbicula*. The collector mayfly Ephemeroptera–*Baetis* was the main contributor to the group with least dissimilarity (C-1 and C-3) (Table 4). However, when we counted only taxa (EPTO), the average dissimilarity percentages of communities between individual channels were lower than the percentages of communities of all taxa (Table 4). The EPTO communities were not significantly different between individual channels (average dissimilarity percentages ranged from 4% to 63%).

Additionally, we considered channels in different land-use groups, such as urban channels, rural channels and agricultural channels, using SIMPER and ANOSIM analysis to see the average percentage of dissimilarity among different land-use groups. The highest average dissimilarity percentages were found between urban channels and rural channels, at 73%, which is the same result found from the comparison of individual channels (C-5 and C-6). Channels in the rural area and channels in the agricultural area showed similar communities (45% dissimilarity) (Table 5).

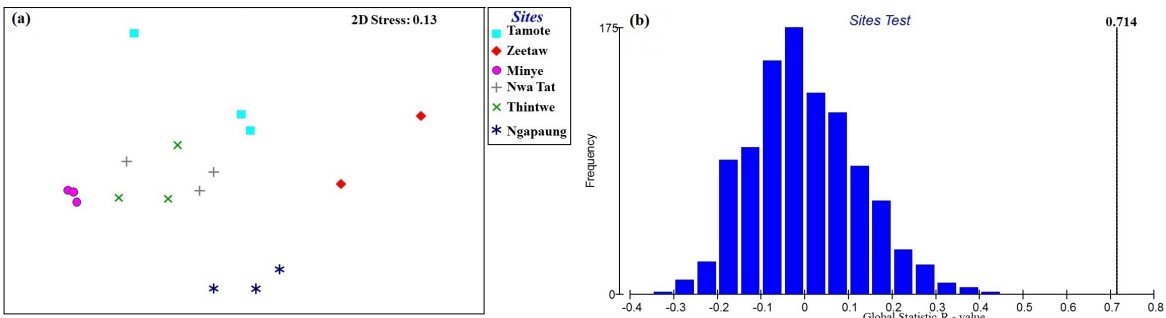

**Figure 5.** (**a**): Dimensionless MDS ordination plot of morpho-species contributions among six channels based on the Bray–Curtis similarity matrix of square root transformation. (**b**): ANOSIM plot for morpho-species contribution similarity among six channels.

**Table 4.** Average dissimilarity percentages among individual channels with their highest species contribution and percentage and functional feeding groups (FFG) (L: larvae, sp.: species) using the morpho-species contributions of all taxa and EPTO by an analysis of similarity percentages (SIMPER)

| Channels | | Average Dissimilarity [%] | | Highest Species Contribution | | Respective FFG | | Contribution [%] | |
|---|---|---|---|---|---|---|---|---|---|
| | | **All Taxa** | **EPTO** | **All Taxa** | **EPTO** | **All Taxa** | **EPTO** | **All Taxa** | **EPTO** |
| C-1 | C-2 | 62 | 33 | Dytiscidae | *Asionurus* sp. 1 | Predator | Filterer | 27 | 34 |
| C-2 | C-3 | 66 | 36 | Dytiscidae | *Baetis* sp. 1 | Predator | Collector | 30 | 38 |
| C-3 | C-4 | 58 | 41 | *Caenis* sp. 1 | *Asionurus* sp. 1 | Collector | Filterer | 17 | 31 |
| C-3 | C-1 | 40 | 33 | *Baetis* sp. 1 | *Baetis* sp. 1 | Collector | Collector | 23 | 35 |
| C-4 | C-5 | 74 | 62 | *Corbicula* | *Asionurus* sp. 1 | Filterer | Filterer | 22 | 35 |
| C-4 | C-2 | 76 | 48 | Dytiscidae | *Asionurus* sp. 1 | Predator | Filterer | 21 | 36 |
| C-4 | C-1 | 58 | 33 | *Caenis* sp. 1 | *Baetis* sp. 1 | Collector | Collector | 16 | 25 |
| C-5 | C-6 | 91 | 4 | *Corbicula* | *Caenis* sp. 1 | Filterer | Collector | 34 | 100 |
| C-5 | C-3 | 91 | 40 | *Corbicula* | *Baetis* sp. 1 | Filterer | Collector | 30 | 48 |
| C-5 | C-2 | 81 | 37 | *Corbicula* | *Gomphidae* sp. 1 | Filterer | Predator | 30 | 48 |
| C-5 | C-1 | 77 | 47 | *Corbicula* | *Asionurus* sp. 1 | Filterer | Filterer | 25 | 32 |
| C-6 | C-4 | 74 | 63 | Thiaridae | *Asionurus* sp. 1 | Grazer | Filterer | 16 | 35 |
| C-6 | C-3 | 50 | 43 | *Baetis* sp. 1 | *Baetis* sp. 1 | Collector | Collector | 26 | 46 |
| C-6 | C-2 | 79 | 40 | Dytiscidae | *Gomphidae* sp. 1 | Predator | Predator | 29 | 45 |
| C-6 | C-1 | 54 | 50 | Elmidae (L) | *Asionurus* sp. 1 | Collector | Filterer | 19 | 31 |

**Table 5.** Results of an analysis of similarity percentages (SIMPER) (average dissimilarity) and one-way analysis of similarity (ANOSIM) (Global statistic R, significant level of statistic, *p*-value) for the comparison of macroinvertebrate communities between channel groups with different land uses based on the functional feeding groups (FFG) of morpho-species contribution.

| Water bodies | Variable | SIMPER Average Dissimilarity | ANOSIM | |
|---|---|---|---|---|
| | | [%] | **R-Statistic** | ***p*-Value** |
| Urban channels–rural channel | | 74 | 0.938 | 0.018 |
| Urban channels–agricultural channels | FFG | 54 | 0.389 | 0.009 |
| Rural channel–agricultural channels | | 45 | 0.221 | 0.011 |

Note: Urban channels are C-4 and C-5; the Rural channel is C-6; Agricultural channels are C-1, C-2, and C-3.

### 3.5. Ecological Water Quality Condition of the River and Channels

All sampling sites in both the river and irrigation channels showed fair ecological conditions, with an average ASPT index ranging from 3.0 to 6.2 among channels and from 4.14 to 4.86 at river sites (Figure 6a).

The statistical analysis of the Bray–Curtis similarity and MDS of the calculated ecological water quality index between the river and channels showed a very low stress value (0.05), indicating that the ordination space between samples exhibited a near-perfect representation [28]. Although the magnitude of the calculated ecological water quality indices was different between different water bodies, there was an overlap between the rank order correlation distance among samples (Figure 6b). A one-way similarity ANOSIM with 999 permutations also confirmed that the variety of sites based on the indices was not significantly different between the river and channels.

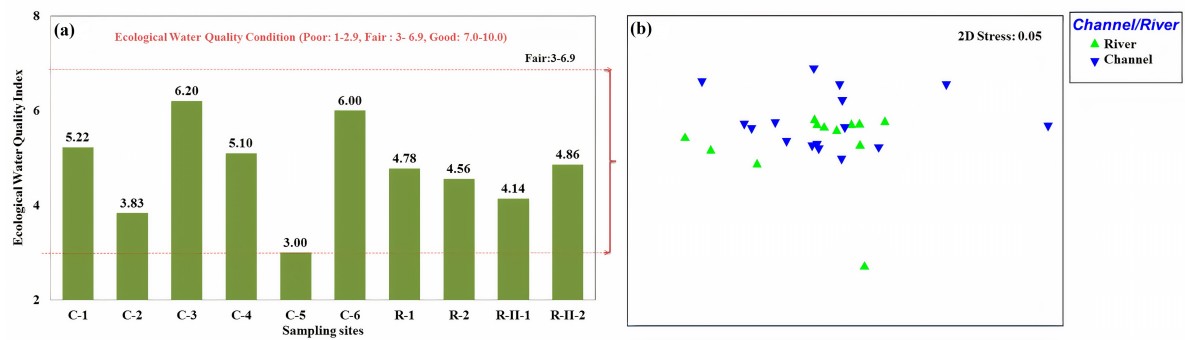

**Figure 6.** (**a**) Calculated ecological water quality indices on the river and channels using the full list of identified families from three composite samples at each site; (**b**) dimensionless MDS ordination plot for the comparison of indices between the river and channels.

## 4. Discussion

### 4.1. Objectives

This study presents pioneering work on the aquatic ecosystems of irrigation channels in Myanmar. Our objective was to investigate the aquatic ecosystem of human-made irrigation channels based on macroinvertebrate communities and to test the differences of these communities to those of the natural river. A number of methods were used: the lowest taxonomic level (morpho-species data), EPTO taxa, FFG comparisons and a rapid biomonitoring method (family level taxonomy) in both river and irrigation channels.

### 4.2. Water Quality

In situ water quality values at all sites of both the river and channels were within acceptable irrigation water quality standards (pH 6.5-8.4, EC < 700 $\mu$S cm$^{-1}$; [29]) but not suitable for household use, as the suspended sediment concentration was quite high (>30 NTU) at the time of testing. The electrical conductivity in channels was also within the normal freshwater aquatic life criteria range (150–500 $\mu$S cm$^{-1}$), as discussed by [20]. EC levels in the river were a little higher than the normal range (546–550 $\mu$S cm$^{-1}$). Small changes in EC level such as those found in our study (546–550 $\mu$S cm$^{-1}$) are not relevant to changes in the macroinvertebrate assemblage [30].

Kartikasari [18] noted that the spot sampling of physico-chemical water quality measurement does not fully capture water quality fluctuations. However, our collected physico-chemical water quality in the Zawgyi River did not show significant changes between the first sampling occasion and the second. Therefore, while not accurately representing the fluctuations that may occur during storms, our findings can be considered representative for baseflow water quality conditions in the dry season.

Additionally, during the field sampling period, we performed a participatory monitoring interview project in our study area [31]. We interviewed inhabitants from two villages near channel C-6 and children from two private high schools in Kyaukse city (near C-4 and C-5). All interviewees complained about pollution and diseases such as cholera but did not mention dramatic water quality changes over the seasons. Generally, the collected physico-chemical water quality variables showed that the water quality was within acceptable standards for irrigation water and freshwater aquatic life; however, it was not acceptable as drinking water without adequate treatment.

Notably, the most sensitive group, Plecoptera, which requires cold and unpolluted freshwater with a high dissolved oxygen level [22], was not recorded at any site in both water body types, suggesting that all sites were impacted by modified water quality. The lack of this family could be related to the nutrient composition of water associated with the direct discharge of household hazardous waste disposal into both river and channels. This corresponded to the situation observed during our field visit. Indeed, we found several hard rubbish waste disposal areas near the river bank and channels

during our sampling collection, but we could not measure the nutrient or other pollutant composition of organic and inorganic materials in this current study.

*4.3. Comparison of Macroinvertebrates Community between River and Channels*

Although the comparison between the natural river and human-made irrigation channels highlighted significantly different taxonomic assemblages of all aquatic taxa (highest taxonomic level of identification; i.e., morpho-species level), the difference between these two water bodies decreased from 80% to 64% when we used only EPTO taxa, which includes the more sensitive groups. The comparison is based on a significantly reduced data set with only four species in the rivers and seven in the channels, with a high abundance of the tolerant mayfly *Caenis* and absence of *Asionurus* in the river.

FFG compositions still support differences between the river and channels, in that collectors and grazers were more present in rivers while filterers were found more often in channels. The higher diversity of collectors in the river is consistent with the study presented in [21], which found that collectors were most abundant in the downstream reach of a forest–agriculture–urban river where high human inputs into the watercourses occurred. The presence of grazers in the river shows that there were ideal conditions to produce algae and biofilm [20]. Filterers in the channels suggest that the substrate is mainly silty/sandy sediments, as the number of filterers has a positive correlation with the total dissolved solids within the water body [32] and a flowing current to provide natural organic matter.

Among the collected FFGs, there was a lack of shredders at all sampling sites at the time of sampling. Dudgeon [22] states that there are a lack of shredders or a relatively low proportion of shredders in tropical Asian streams. For example, shredders encompass not more than 8.8% of the communities in the streams in Hong Kong. The lack of shredders in our study could be related to the absence of leaf input for decomposed terrestrial leaf litter in the Zawgyi River and irrigation channels.

This difference in FFGs might be related to the different substrates between river and channels (rocky and earth, respectively), which could support different food sources, with aquatic macrophytes and algae in the river and sediments and detritus in the channels. The input of anthropogenic pressure and agricultural activities on water bodies also create changes in FFG variation [21].

Additionally, the different communities between the river and channels might reflect the different habitat availability and hydrological regimes. Irrigation channels are not streams but water courses that do not mimic the flow regimes of streams but rather ensure flows in constructed channels and thus do not create stream-like habitat diversity. The channels are subjected to dry or low water levels (ranging from 0.15 to 0.5 m) during the late-season of crop development, enabling fresh harvest or dry harvest based on the particular crop, whereas the river fluctuates less throughout the year, with a constant baseflow and with wet season fluctuations above this. For example, Kartikasari [18] reported that the dominant taxa (*Melanoides tuberculata*–Thiaridae) in irrigation channels show whether dry periods occur or not. This species of gastropod snail has an operculum, enabling it to seal its shell and withstand dry periods. We also found a high abundance of these Thiaridae snails in almost all channels, except C-2 and C-5; the Thiaridae snail was also found in the Zawgyi River, although only a few individuals were present.

*4.4. Aquatic Ecosystem of Irrigation Channels*

The macroinvertebrate communities in the irrigation channels were not homogenous. The highest dissimilarity was found in channels C-5 and C-6. This is logical in the case of the different anthropogenic activities within agricultural or urban areas, as discussed in [13].

This holds particularly for filterers, which feed mainly on detritus and fine organic matter in the water and are usually buried in the sediment bed of the waterbody [33] and were abundant in C-5. In the C-6 channel, grazers (Thiaridae) were highly abundant. Grazers feed on soft sediments, as at channel C-6 where they feed in the mud and are effectively grazing in the mud [34]. The difference in

the physical structure of channels may affect the flow velocity, which relates to the water–sediment interface within the water column [12]. However, the most abundantly found taxa in all channels were the tolerant mayflies *Baetis*, *Asionurus* and *Caenis*. Therefore, EPTO taxa did not show significant differences among individual channels (Table 5).

### 4.5. Ecological Water Quality Index

All sampling sites showed fair ecological water quality conditions (Figure 6a). Different assemblages of families gave different ecological water quality indices based on the ASPT and taxa richness at each site, but all sites still ranked as "fair". This result—the same fair condition—indicates that all the sampling sites are impacted by human disturbance in both water bodies (e.g., direct household waste disposal, organic pollutants) but show different physical and chemical characteristics (e.g., substrate, flow velocity, wetted width and hydrology). Although this new water index method for Myanmar did not show significant differences between sites—unlike the morpho-species, EPTO and FFG indices—it did show that human activities in the lower section of the Zawgyi River resulted in impaired water quality. Therefore, it becomes clear that the Myanmar based biomonitoring index, which is still under development [23], should be extended to improve its sensitivity.

### 4.6. Challenges and Citizen Science Potential

The lack of detailed knowledge about aquatic biodiversity in Myanmar makes our study challenging and hinders the extrapolation of our findings to other studies in the same region. Additionally, the limited availability of long-term instantaneous physico-chemical data for smaller water bodies in Myanmar is restrictive. Our research, then, aims to fill a critical knowledge gap, which is necessary to manage Myanmar's freshwater resources for sustainable development. No macroinvertebrate studies focusing on irrigation channels have been published in Myanmar. However, recently, Ko et al. [23] studied macroinvertebrate communities upstream of hydropower dams in the Myitnge and Chaungmagyi Rivers in the Eastern part of Myanmar. They found fair to good ecological water quality using three different international biomonitoring indexing methods (The South African Scoring System: miniSASS, Australian Waterwatch and Asia Foundation methods). However, our current study is also the first study of artificially constructed water courses in Myanmar. The Myanmar-based biomonitoring index method [23] showed that the sampling sites are all of low/fair ecological water quality, but we had no high-quality sites for comparison. However, at higher levels of taxonomic identification, our study showed the differences in communities among sampling sites, because we considered morpho-species and abundance together. The index method, a family-based method, is a reduction of data information, but this does not remove its value in water quality monitoring. Therefore, the preliminary, Myanmar-based biomonitoring index is a potential indicator for water resource research, particularly in the area of citizen science. However, when we apply the index method as a citizen science approach, there will always be difficulties in terms of data accuracy and reliability; however, there are gains from helping local communities to monitor their water quality using macroinvertebrates. During our field sampling period, we tested indexing methods in the field with Grade 6 and 8 students from a high school near channels C-4. Most of the students could detect water quality well when they used the index method with an order-based method (miniSASS: http://www.minisass.org/en/) rather then a family-based method (Asia Foundation, which formed the basis of the Myanmar-based biomonitoring index) during their field visit. Ko et al. [23] also recommended the use of a much-easier order-based method such as miniSASS, which is more suitable for non-professional citizen participation approaches. Such a citizen science approach could increase awareness of river health as well as influencing local water management decisions.

## 5. Conclusions

Irrigation channels are small, artificial water bodies that not only play an essential role in social-economic activities but also provide a habitat for aquatic fauna and flora in agricultural countries such as Myanmar. However, small water bodies are often overlooked in aquatic ecological studies compared to the larger natural and modified rivers. We sampled both river and irrigation channels to determine how the macroinvertebrate communities altered between the Zawgyi River and human-made irrigation channels. Our findings were that the diversity of macroinvertebrates was low in both water bodies, which we linked to the (household) pollution of these environments. Thus, in the absence of impacted streams, it is likely that irrigation channel communities will merely be similar to impacted stream communities; i.e., reduced biodiversity and the selection of more tolerant taxa. We also illustrated the use of macroinvertebrate communities to detect ecological water quality in different water bodies in Myanmar, which could have a potential role in a program of participatory monitoring by local people and management authorities.

Our study is a pioneering study with a limited dataset which aimed at the assessment of irrigation channel ecosystems in Myanmar using aquatic macroinvertebrate assemblages. Further aquatic ecology research in Myanmar on other natural and semi-natural water bodies is recommended. We also recommend that researchers should monitor and document irrigation water quality, with a focus on nutrient loads, in further studies. Molecular analyses using DNA could be used in the future to verify morphological identification and to separate closely related species, but this was considered to be beyond the scope of this ecological study as it aimed to support community-based monitoring by local people. We suggest that future research should focus on extending the spatial scale of macroinvetebrate sampling with a multi-year sampling period with increased replicates (five) at each site to fill the knowledge gap of the biodiversity of small water bodies in Myanmar.

**Author Contributions:** Conceptualization, N.T.K., P.S., J.C., M.R. and T.B.; methodology, N.T.K., P.S., J.C., M.R. and T.B.; software and formal analysis, N.T.K. and P.S.; validation, N.T.K. and P.S.; writing—original draft preparation, N.T.K.; writing—review and editing, P.S., J.C., M.R. and T.B. All authors have read and agreed to the published version of the manuscript.

**Funding:** This research was funded by The Netherlands Fellowship Program (NFP) for a four-year PhD study. The APC was funded by Delft University of Technology, The Netherlands.

**Acknowledgments:** The authors would like to acknowledge the local people from villages and students from high schools who participated in the field sampling campaign of this research. Thanks also go to the Department of Irrigation, Kyaukse for the supporting information about Myogyi dam, weirs and irrigation channels.

**Conflicts of Interest:** All authors declare they have no conflict of interest in this research.

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
