# Peer review of "Aquatic Macroinvertebrate Indicators in the Zawgyi Irrigation Channels and a River in the Central Dry Zone of Myanmar"

_sustainability, doi:10.3390/su12218788_

Round 1

Reviewer 1 Report

Good job improving the manuscript.

Author Response

On behalf of my co-authors and me, thank you, the reviewer for the nice words.

Reviewer 2 Report

Authors made most of the changes suggested during first revision. However they overlooked some of them, although the changes requested were few. I recommend to pay more attention.

Below I re-paste the changes still to be made:

Table 1:

  • Please check and uniform the terms in the legend: adult (singular) vs. pupae (plural).
  • Hemiptera Micronectidae are not Predators. Their diet is still unknown for most the species, however some species eat both algae (mainly) and annelids and bacteria.

  • For some taxa “sp” is not reported ( Macrobrachium).

Author Response

On behalf of my co-authors and me, thank you, the reviewer for the nice words.

This manuscript is a resubmission of an earlier submission. The following is a list of the peer review reports and author responses from that submission.

Round 1

Reviewer 1 Report

This manuscript explores the macroinvertebrate communities of irrigation channels in a river basin in Myanmar and compares them to the macroinvertebrate communities of the river they drain to. The study goal is interesting and could help to fill a knowledge gap on the biological communities inhabiting irrigation channels and on the macroinvertebrate communities in Myanmar. However, I have also major concerns about the study mainly due to the low number of samples and study sites included.

  1. The major limitation of the manuscript is that it only includes 10 macroinvertebrate samples, with 3 sub-samples/replicates each. Only 8 sites were sampled, and the 6 sites located at irrigation channels were sampled only once. Sub-samples/replicates appear combined as single samples in some analysis (e.g. Table 1) or included as “independent” samples in other analysis (e.g. Figure 3). I consider that low number of samples insufficient to achieve the study goals (i.e. to describe the aquatic ecosystem of irrigation channels using aquatic macroinvertebrates communities and to compare them to river ecosystems).

  1. Choice of mesh size is a critical point for macroinvertebrate sampling, because finer mesh will capture more organisms. A mesh size of 500 μm is usually considered adequate, although some researchers and monitoring programs use finer mesh sizes (e.g. 400 μm, 250 μm). See reference 8 in the manuscript for a deeper discussion on that topic. In this work, it was unclear which mesh size was used (line 118: “a sweep-sample technique with Nylon nets (500 MU and 1,000 MU mesh)”. In fact, the text indicates that two different mesh sizes were used, one mesh size of 1000 μm, which is uncommon for macroinvertebrate sampling because it would not capture small macroinvertebrates. Were different mesh sizes used for different sites? When and where was the 1000 μm mesh size used?

  1. There are some inconsistencies in the manuscript. For instance, the text (lines 177-178) indicates that site C-6 presented 3 different taxa (“the lowest richness was three taxa in site C-6 (Table 1).” However, Table 1 indicates that taxa richness in C-6 was 2, but at the same time, Table 1 shows abundances for 4 different taxa in site C-6.

  1. Macroinvertebrates are classified in four FFGs: collectors, filterers, grazers and predators. This classification is uncommon because does not include shredders. It should be justified.

  1. Regarding the theory framework, irrigation channels are depicted as essential for human activities and biodiversity. While aquatic species can inhabit irrigation channels, these human structures are not essential for biodiversity. The introduction and discussion should be more critical about the negative impacts of the construction of irrigation channels on nature.

Author Response

We thank the reviewer for the nice words and the important feedback on the limitations of our dataset in relation to the research objective. We agree that our total amount of samples is modest, but our data do show that in human-made irrigation channels the diversity of macroinvertebrates was low mainly due to pollution of these environments and we show the opportunities of the use of macroinvertebrate communities to detect ecological water quality. We agree with the reviewer more work is required and we hope our findings will stimulate such work. We made this clear by a slight modification of the research objective and conclusions.

Sincerely,

Nyein Thandar Ko

Reviewer 2 Report

I immediately noticed that diversity in both irrigation channels and rivers is very low, considered the original richness of such environments in SE Asia. Although pollution probably played the main role, I suspect it is in part due also to under-sampling (due also to standardized sampling methods) and, maybe, at least in some cases, overlooking of some species (under the same name as morpho-species). Therefore I suggest at least to avoid to write that these water bodies “support a relative diverse aquatic community” (Conclusions), but instead to stress that diversity is low mainly due to pollution of these environments.

Some comments concerning Table 1:

- Please check the terms in the legend: adult (singular) vs. pupae (plural), specie = species.

- Sometimes between “sp” and numbers there is a space, sometimes not (sp1 vs. sp 1).

- Hemiptera Micronectidae are not Predators. Their diet is still unknown for most the species, however some species eat both algae (mainly) and annelids and bacteria.

- For some taxa “sp” is not reported (see e.g. Macrobrachium, Hemiptera Micronectidae or Neuroptera Sisyridae)

Some comments concerning Table 4:

- Since the presence of the letter L for larvae of Elmidae, I suggest to repeat this abbreviation also in the legend of Table 4.

- In one case “Gomphidae sp 1” in another “Gomphidae sp”. Please uniform them.

Author Response

We thank the reviewer for his time and constructive review.

Sincerely,

Nyein Thandar Ko

Reviewer 3 Report

The paper  “Aquatic Macroinvertebrate Indicators in Irrigation Channels and River in Myanmar” conducted macroinvertebrate sampling with 500 and 1000 um sieves/nets and used the resulting data along with water quality measurements and interviews with local people to determine the ecological integrity of the channels versus the river. Although the approach was comprehensive there are some concerns with the sampling repeats. They stated that they were unable to do a repeat sampling for the channels. It is not clear how this impacted their results. The authors should address this in the discussion. they state that these are pioneer studies and might be better suited for a short communication based on the low sampling repeats. A multi year sampling effort might reveal different ecological integrity results which might be impacted by changing biotic and abiotic factors such as precipitation, pollution etc..

To improve the overall value of the paper I would add to the introduction and discussion on the important types of macroinvertebrates that can show ecological integrity. They never mention Arachnids which are an important bioindicator (water mites). This might be a result of their mesh size and they should consider using a 250 um mesh size. They could also include photographs of the macroinvertebrates that they collected. This would add value to the paper since Myanmar is a biodiverse hot spot. Since they put the specimens in ethanol they could do DNA barcoding on the samples that could identify potentially new species. This could be presented as a future direction. They should increase the description of the taxonomy used to identify the organisms. This is important for the community in order to repeat the identifications.

The paper should be revised and could be published after the recommended changes.

Author Response

We thank the reviewer for the nice words and the very constructive review. The reviewer has a good point here, indeed we could not present a multiyear sampling effort in this current study. However, we remain confident that we showed the value of our methods and see clear communities in both human made irrigation channels, but agree that more sampling periods should be done to get the different ecological integrity results. But, we also believe this work is an important step into that direction in Myanmar where there is lack of knowledge about the use of aquatic macroinvertebrate communities on ecological water quality of both artificial and natural water bodies. Therefore, we stated that our current study is a pioneer study and recommended for further work with spatial scale of macroinvertebrates sampling work.

Sincerely,

Nyein Thandar Ko

Round 2

Reviewer 1 Report

I congratulate the authors for improving the manuscript. Some of the major comments I highlighted in my first review has been addressed.

As highlighted in my first review, the major limitation of the manuscript is that it only includes 10 macroinvertebrate samples, with 3 sub-samples/replicates each. Only 8 sites were sampled, and the 6 sites located at irrigation channels were sampled only once. This limitation has not been improved. I understand the study design cannot be changed, but other data from published sources can be included in the discussion, for instance, in sections 4.3 and 4.5.

In this vein, part of the manuscript (including the title, and specially the introduction) sounds like the results obtained from this limited samples are representative of the whole country of Myanmar and to all the irrigation channels in the country. This should be corrected.

I also included in my first review that sub-samples/replicates appear combined as single samples in some analysis (e.g. Table 1) or included as “independent” samples in other analysis (e.g. Figure 3). This has not been justified in the manuscript.

Detailed comments can be found below:

Title

The title of the article is too wide. The manuscript only includes data of one river (Zawgyi River) and six associated channels.

Abstract

Lines 1-3. Some of these services are NOT dependent on a healthy ecosystem. For instance, hydropower does not depend on a healthy river, in contrast hydropower negatively impact on the health of rivers.

Lines 3-4. Irrigation channels are NOT essential, are currently used for daily water use, maybe important for the current farming model, but not essential.

Line 8. PRIMER v6 is a software not an analysis. Please, include here the main analysis.

Line 13-16. This is contradictory with the previous lines. How the biomonitoring index method could be applied to human-made channels if no significant differences were found between river and channels in this study case?

Introduction

Line 31-32. Artificial or human-made smaller water bodies are NOT essential components for human society neither for aquatic flora nor for fauna communities.

Line 61: there are five FFGs, shredders are missing there. If there were not shredders in the analyzed samples, that can be explained in the methods/results.

Lines 72-74: The first sentence seems to have something lacking, I suppose the goal is to compare the irrigation channels with the river. Only one river was sampled.

Materials and Methods

Line 82 and line 95. Why were these sites selected and no other sites?

Results

Line 234. Spelling mistake, it should be SIMPER

Figure 1. Please, mark the flow direction.

Figure 2 and Figure 6a. I suggest to improve the figures using just solid colors, no patterns.

Discussion

Lines 300-304. I really appreciate this improvement.

Line 316. The species name should be in italic.

Line 324. I think “Leslie et al.” is missing here.

Line 333-343. Because the manuscript presents data from only 8 sites. Can these results be compared with other results from other rivers? What does “fair” means in other river sites? How impacted are the sites elsewhere with an index <3?

Line 375 and elsewhere. The causes of the low diversity of macroinvertebrates in both water bodies were not demonstrated in this study. It can be pollution, but can also be channel modification, hydrology alteration…

Reviewer 3 Report

I am assuming the authors have no pictures of the macroinvertebrates sampled. They did not address this. They also should indicate if they stored the samples in ethanol which could then be used for DNA barcoding in the future. This would assist with easier biomonitoring of the region especially if there are cryptic or difficult to identify organisms.

Please cite the following paper which reviews, amongst others, a paper you have cited in your ms.

Use of Aquatic Biota to Detect Ecological Changes in
Freshwater: Current Status and Future Directions
José Maria Santos * and Maria Teresa Ferreira. Published in Water. 2020.